# Genetic association of serum lipids and lipid-modifying targets with endometriosis: Trans-ethnic Mendelian-randomization and mediation analysis

**Hongling Zhang[1], Yawei Fan[2]\*, Huijun Li[3], Xiaoqing Feng[1], Daoyuan Yue[3]**

**1** Gynecology Department of Tongji Hospital, Tongji Medical-College, HUST, Wuhan, Hubei, China,
**2** General Surgery of Tongji Hospital, Tongji Medical-College, HUST, Wuhan, Hubei, China, **3** The
Laboratory Medicine Department of Tongji Hospital, Tongji Medical-College, HUST, Wuhan, Hubei, China

\* yzk221523@163.com

## Abstract

org/10.1371/journal.pone.0301752

AUSTRALIA

**Data Availability Statement:** All relevant data are
within the paper.

**Funding:** The author(s) received no specific
funding for this work.

### Background

Prior observational research identified dyslipidemia as a risk factor for endometriosis (EMS)
but the causal relationship remains unestablished due to inherent study limitations.

### Methods

Genome-wide association study data for high-density lipoprotein cholesterol (HDL-C), low-
density lipoprotein cholesterol (LDL-C), triglycerides (TG), and total cholesterol (TC) from
European (EUR) and East Asian (EAS) ancestries were sourced from the Global Lipids
Genetics Consortium. Multi-ancestry EMS data came from various datasets. Univariable
Mendelian randomization (MR) examined causal links between serum lipids and EMS. Mul-
tivariable and mediation MR explored the influence of seven confounding factors and media-
tors. Drug-target MR investigates the association between lipid-lowering target genes
identified in positive results and EMS. The primary method was inverse-variance weighted
(IVW), with replication datasets and meta-analyses reinforcing causal associations. Sensi-
tivity analyses included false discovery rate (FDR) correction, causal analysis using sum-
mary effect estimates (CAUSE), and colocalization analysis.

### Results

IVW analysis in EUR ancestry showed a significant causal association between TG and
increased EMS risk (OR = 1.112, 95% CI 1.033–1.198, $P = 5.03 \times 10^{-3}$, $P_{FDR} = 0.03$), sup-
ported by replication and meta-analyses. CAUSE analysis confirmed unbiased results ($P <$
0.05). Multivariable and mediation MR revealed that systolic blood pressure (Mediation
effect: 7.52%, $P = 0.02$) and total testosterone (Mediation effect: 10.79%, $P = 0.01$) partly
mediated this relationship. No causal links were found between other lipid traits and EMS ($P$
> 0.05 & $P_{FDR} > 0.05$). In EAS ancestry, no causal relationships with EMS were detected ($P$
> 0.05 & $P_{FDR} > 0.05$). Drug-target MR indicated suggestive evidence for the influence of

**Competing interests:** The authors have declared that no competing interests exist.

ANGPTL3 on EMS mediated through TG (OR = 0.798, 95% CI 0.670–0.951, $P$ = 0.01, $P_{FDR}$ = 0.04, PP.H4 = 0.85%).

## Conclusions

This MR study in EUR ancestry indicated an increased EMS risk with higher serum TG levels.

## Introduction

Endometriosis (EMS) is one of the most common disorders in women of reproductive age, characterized by the presence of endometrial tissue (glands and stroma) above and outside the uterine cavity, affecting up to 6%-10% of women [1, 2]. Women with EMS often experience painful menstruation, pelvic pain, infertility, or difficulty conceiving, which not only impacts their physical health but also results in significant economic burdens including medical costs and loss of work capacity [3]. EMS is characterized by estrogen dependence, and numerous studies have suggested that it may be associated with genetics, immune system dysregulation, hormonal imbalances, and environmental factors [4, 5], yet the specific pathological mechanisms require further clarification. Given the complex etiology and challenging prevention of endometriosis, there is an urgent need to identify its potential risk factors.

Lipids are not only crucial components of biological structures but also play key roles in various signaling pathways, significantly impacting cellular functions. Abnormalities in lipid metabolism can affect hormonal balance and may indirectly influence the growth and function of the endometrium by modulating inflammatory responses and immune functions [6]. A substantial body of evidence from previous studies suggests a correlation between serum lipid levels and an increased incidence of EMS [7–11]. However, most of these observed associations are based on observational designs, and their causal relationships have not been fully established.

Randomized controlled trials (RCTs) are theoretically ideal for providing robust causal evidence, yet they are often impractical due to their high costs and extensive time and resource requirements. Mendelian Randomization (MR), an alternative analytical approach resembling RCTs, leverages genetic variants identified in genome-wide association studies (GWAS) as instrumental variables (IVs) to determine the causal effects of risk factors (or exposures) on outcomes [12]. MR capitalizes on the random allocation of alleles related to exposures at conception, thereby bypassing confounders and reverse causation problems frequently encountered in conventional epidemiological studies [13]. This study aimed to employ MR analysis to investigate the genetic susceptibility of high-density lipoprotein cholesterol (HDL-C), low-density lipoprotein cholesterol (LDL-C), triglycerides (TG), and total cholesterol (TC) to the risk phenotype of EMS. Furthermore, this study delved into blood lipid levels that have a significant causal link with EMS, identifying lipid-lowering gene targets. The objective was to clarify the causal nexus between lipid levels influenced by these gene targets and EMS.

## Materials and methods

### Study design

This research utilized primary datasets from publicly accessible GWAS to investigate potential causal relationships between specific exposures and outcomes. A diverse array of advanced

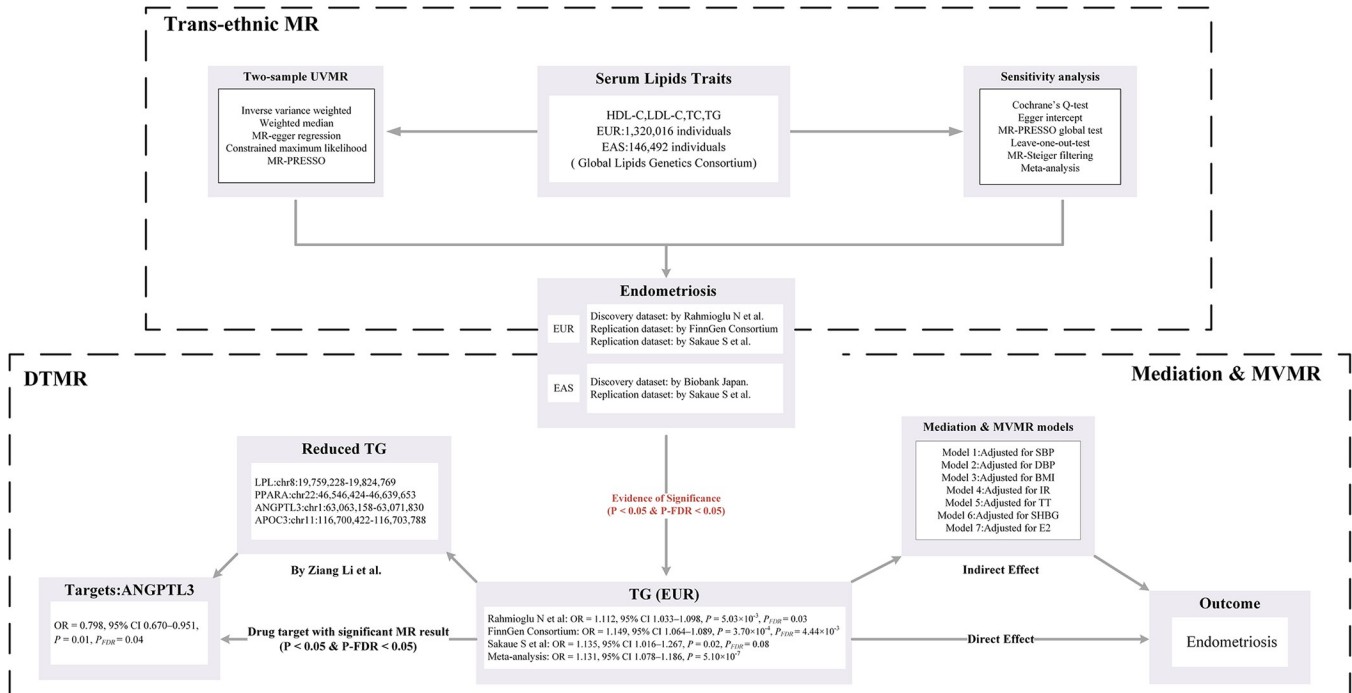

**Fig 1. Overview of research design and analysis strategy.** MR, Mendelian randomization; MVMR, Multivariate MR; UVMR, Univariable MR; DTMR, Drug-target MR; BMI: Body Mass Index; SNP, Single Nucleotide Polymorphism; MR-PRESSO, MR Pleiotropy Residual Sum and Outlier; LDL-C, Low-Density Lipoprotein Cholesterol; HDL-C, High Density Lipoprotein Cholesterol; TG, Triglyceride; IR, Insulin resistance; TC, Total cholesterol; TT, Total Testosterone; SHBG, Sex hormone binding globulin; EUR, European; EAS, East Asian; E2, Oestradiol; DPB, Diastolic blood pressure; SBP, Systolic blood pressure; ANGPTL3, angiopoietin-like 3; FDR, false discovery rate.

analytical methods was employed, encompassing univariable MR (UVMR), multivariable MR (MVMR), and mediation MR analysis. The criteria for selecting IVs for these exposures were stringent: (i) the chosen genetic markers, serving as IVs, must exhibit a strong association with the exposure; (ii) these markers should not be associated with potential confounding variables; and (iii) the impact of the genetic variants on the outcome should be solely through the exposure, precluding other mechanisms [14]. Additionally, in light of the positive results of the study, drug-target MR (DTMR) analysis was conducted, focusing on lipid-lowering gene targets identified through serum lipid phenotypes. The MR methodology employed is depicted in **Fig 1**. Detailed summary statistics from these datasets are methodically outlined in **Table 1.**

## Selection of genetic instrumental variables

To maintain rigorous and precise MR evaluations, our study implemented strict criteria for selecting single nucleotide polymorphisms (SNPs) as IVs:

(i) SNPs designated as IVs were mandated to exhibit a genome-wide significant association with the designated exposure ($P < 5×10^{-8}$). (ii) The chosen SNPs underwent comprehensive analysis, utilizing the East Asian and European populations from the 1000 Genomes Phase 3 reference panel, to exclude associations with potential confounders. By setting strict LD parameters ($r^2 < 0.001$, clumping distance = 10,000 kb), we ensured the uniqueness of our SNP selection and minimized LD biases, catering to the genetic diversity of these populations. (iii) The effectiveness of each SNP as an IV was evaluated using an F-statistic, simplified for a single SNP as $F = ((n-k-1)/k)\ (R^2/(1-R^2))$, where $R^2$ is the proportion of variance in the exposure explained by the SNP, n is the sample size, and k = 1 reflecting the individual SNP

**Table 1. Detailed information on data sources.**

| Phenotype | Ieu/EBI ID | Ref | Consortium | Ancestry | Participants |
|---|---|---|---|---|---|
| **Exposure** | | | | | |
| HDL-C | GCST90239652 | 34887591 | GLGC | EUR | 1,320,016 individuals |
| LDL-C | GCST90239658 | 34887591 | GLGC | EUR | 1,320,016 individuals |
| TC | GCST90239676 | 34887591 | GLGC | EUR | 1,320,016 individuals |
| TG | GCST90239664 | 34887591 | GLGC | EUR | 1,320,016 individuals |
| HDL-C | GCST90239651 | 34887591 | GLGC | EAS | 146,492 individuals |
| LDL-C | GCST90239657 | 34887591 | GLGC | EAS | 146,492 individuals |
| TC | GCST90239675 | 34887591 | GLGC | EAS | 146,492 individuals |
| TG | GCST90239663 | 34887591 | GLGC | EAS | 146,492 individuals |
| **Discovery dataset** | | | | | |
| EMS | GCST90269970 | 36914876 | Rahmioglu N | EUR | 21,779 cases / 449,087 controls |
| EMS | bbj-a-114 | 28189464 | BBJ | EAS | 734 cases / 102,372 controls |
| **Replication dataset** | | | | | |
| EMS | NA | 36653562 | FinnGen | EUR | 15,088 cases / 107,564 controls |
| EMS | ebi-a-GCST90018839 | 34594039 | Sakaue S | EUR | 4,511 cases / 227,260 controls |
| EMS | ebi-a-GCST90018619 | 34594039 | Sakaue S | EAS | 1,786 cases / 80,975 controls |
| **Adjustment of the model** | | | | | |
| SBP | ieu-b-38 | 30224653 | ICBP | EUR | 757,601 individuals |
| DBP | ieu-b-39 | 30224653 | ICBP | EUR | 757,601 individuals |
| BMI | ieu-b-40 | 30124842 | GIANT | EUR | 681,275 individuals |
| IR | ieu-b-118 | 20081858 | MAGIC | EUR | 37,037 individuals |
| TT | ieu-b-4864 | 36402876 | UKB | EUR | 199,569 individuals |
| SHBG | ieu-b-4870 | 36402876 | UKB | EUR | 214989 individuals |
| E2 | ieu-b-4872 | 36402876 | UKB | EUR | 53391 individuals |

EMS, endometriosis; MAGIC, Meta-Analyses of Glucose and Insulin-related traits Consortium; BMI, body mass index; GIANT, Genetic Investigation of Anthropometric Traits; TG, triglycerides; LDL-C, Low-Density Lipoprotein Cholesterol; HDL-C, High Density Lipoprotein Cholesterol; GLGC, Global Lipids Genetics Consortium; TC, total cholesterol; UKB, UK Biobank; BBJ, Biobank Japan; ICBP, International Consortium of Blood Pressure; DPB, Diastolic blood pressure; SBP, Systolic blood pressure; IR, insulin resistance; TT, Total Testosterone; SHBG, Sex hormone binding globulin; EUR, European; EAS, East Asian; E2, Oestradiol; NA, not available.

analysis. This approach ensured the robustness of our IVs by excluding any SNPs with an F-statistic below 10, indicative of weak instruments [15]. $R^2$ values for each selected SNP were calculated using the equation $2 \times MAF \times (1-MAF) \times beta^2$, where MAF represents the minor allele frequency. These calculated values were then combined to derive the coefficient critical for the determination of statistical power [16]. (iv) MR-Steiger filtering was applied to further validate our results, removing any variants demonstrating a stronger correlation with the outcome than the exposure [17]. (v) If an SNP is absent in the result dataset, the TwoSampleMR package is utilized to identify the missing SNP. Subsequently, a proxy SNP is selected that exhibits strong linkage disequilibrium ($r^2 > 0.8$) with the principal SNP. (vi) A key criterion for SNP inclusion was that its impact on both the exposure and outcome consistently aligned with the same allele direction.

Based on the findings of this study and insights from previous MR study [18], four triglyceride (TG)-lowering target genes were identified: lipoprotein lipase (LPL), Peroxisome Proliferator-Activated Receptor Alpha (PPARA), angiopoietin-like 3 (ANGPTL3), and Apolipoprotein C-II (APOC3). Employing methodologies used in prior studies [18], SNPs associated with TG levels were screened from the target gene regions (±100 kb from the gene

location), demonstrating genome-wide significance ($P < 5{\times}10^{-8}$). To maximize the instrumental strength of each drug, SNPs utilized as instruments were allowed to exhibit low LD with each other ($r^2 < 0.30$), located within ±100 kb windows from the gene region. The remaining criteria were consistent with those previously described in sections (iii), (iv), (v), and (vi).

## Source of the lipid trait phenotype

The most comprehensive and recent GWAS data on LDL-C, HDL-C, TG, and TC are sourced from the Global Lipids Genetics Consortium (GLGC) [19]. This consortium, led by Graham SE's team, conducted a meta-analysis encompassing five genetic ancestral groups. In this study, the European (EUR) ancestry included data from 146 cohorts comprising 1,320,016 individuals, while the East Asian (EAS) ancestry encompassed 40 cohorts with 146,492 individuals. The GWAS utilized MR-MEGA for meta-analysis of individual cohort files to elucidate the heterogeneity in effect sizes of lipid variations among different ancestral groups. In total, 941 lipid-related loci reaching genome-wide significance ($P < 5{\times}10^{-8}$) were identified, including 355 novel loci from single-ancestry or trans-ancestry analyses.

## Source of endometriosis phenotype

In EUR ancestry, the most comprehensive and recent summary-level GWAS data for EMS in the discovery dataset were derived from a meta-analysis by Rahmioglu N and colleagues [20], encompassing 24 cohorts. This study identified 42 genome-wide significant loci, including 49 distinct association signals, and involved data from 21,779 cases and 449,087 controls across 10 countries. In the replication dataset, data were obtained from the FinnGen consortium's R9 version [21], comprising 15,088 cases and 107,564 controls, and from a GWAS by Sakaue S et al., including 4,511 cases and 227,260 controls [22]. In EAS ancestry, the GWAS data for the Discovery dataset were sourced from BioBank Japan (BBJ), involving 734 cases and 102,372 controls [23]. The replication dataset was derived from Sakaue S et al., including 1,786 cases and 80,975 controls [22].

## Data sources for possible confounders

The study further obtained genetic associations for body mass index (BMI) from the Genetic Investigation of Anthropometric Traits (GIANT) consortium [24]. Data for systolic blood pressure (SBP) and diastolic blood pressure (DBP) were sourced from the International Consortium for Blood Pressure (ICBP) [25], while insulin resistance (IR) data came from the Meta-Analyses of Glucose and Insulin-related traits Consortium (MAGIC) [26]. Additionally, total testosterone (TT), sex hormone-binding globulin (SHBG), and oestradiol (E2) datasets, stratified by sex to ensure specificity to female participants, were all obtained from the UK Biobank (UKB) [27].

## Statistical analyses

### MR analysis

In the MR analysis conducted, the UVMR framework utilized the Wald ratio test for the evaluation of each IV. Concurrently, to establish causal relationships across multiple IVs ($\geq$2), the multiplicative random-effects inverse-variance-weighted (IVW) method was employed. This methodology was further enhanced by incorporating the MR-Egger and weighted median approaches. Within the IVW method, the weighting is aligned with the Wald ratio of each SNP and is inversely related to its variance [28]. The IVW method, encompassing a wide

spectrum of genetic variants, yields consistent and systematic outcomes. The weighted median method remains robust when up to 50% of the information is derived from potentially invalid instrumental variables, allowing for reliable estimation, while the MR-Egger method allows for the identification and adjustment of directional pleiotropic effects, assuming that the instrument-outcome relationship is independent of the exposure [29]. The constrained maximum likelihood (CML) technique was also applied, facilitating comprehensive assessments across a range of genetic variants, and accounting for possible confounders and genetic variation. This approach is particularly effective when dealing with a vast number of genetic variants and confounders, ensuring accurate and robust results [30]. In the DTMR analysis, the primary method employed was the random-effects model version of the IVW approach. For handling multiple comparisons within our analysis, the false discovery rate (FDR) correction method was applied. After FDR adjustment, a *P* -P-value below 0.05 signifies a statistically significant causal relationship. In contrast, a *P* -P-value below 0.05 with an FDR-adjusted *P*-value above 0.05 is considered suggestive but not definitive of such a relationship.

To clarify the direct causal pathways from exposure to outcome, additional multivariable MVMR analyses were undertaken [31]. These analyses were directed at meticulously delineating the direct causal links, differentiating them from the UVMR model. MVMR, as opposed to UVMR which concentrates on a single exposure, considers genetic variations associated with multiple exposures. The initial stage of this analysis involved procuring MR effect estimates for the exposure-to-outcome relations using the IVW method. Subsequently, an MVMR analysis was conducted to evaluate the influence of seven mediators on the outcome, taking into account the characteristics of the exposure. The indirect effects of the exposure were ascertained by calculating the derived estimates for each respective outcome. The culmination of this analysis entailed determining the ratio of the mediation effect to the total effect, thereby shedding light on the proportional impact of these mediators on the overall outcome.

## Colocalization analysis

To ascertain shared causal genetic variants linked to phenotypes elucidated through MR, colocalization analysis was conducted using the Coloc package (version 3.2–1) in R [32]. The genetic variant demonstrating the strongest association with the exposure in the MR analysis, as indicated by the lowest *P*-value, was selected as the reference variant. Variants located within ±100 kb of this reference were included in the analysis. The LD reference panel utilized was based on the 1000 Genomes v3 dataset of European ancestry. For colocalization, a criterion was set where a posterior probability exceeding 0.8 indicated a shared causal variant (posterior probability of hypothesis 4 > 0.8) [33].

## Sensitivity analysis

Within the UVMR analysis, various methodological evaluations were performed. Heterogeneity among the chosen genetic variants was assessed using Cochran's Q test, with a P-value of less than 0.05 indicating significant variability among the SNPs [34]. Directional pleiotropy within the MR framework was investigated via MR-Egger regression [35], where an intercept P-value less than 0.05 implies notable directional pleiotropy, acknowledging the methodological constraints [36]. The MR Pleiotropy Residual Sum and Outlier (MR-PRESSO) approach was employed to detect potential outliers and assess horizontal pleiotropy, which is confirmed if the global P-value is below 0.05 [37]. When unavoidable horizontal pleiotropy was detected by MR-PRESSO, the causal analysis using summary effect estimates (CAUSE) methodology was applied [38], effectively managing both correlated and uncorrelated horizontal pleiotropy within genome-wide datasets [37]. Correlated horizontal pleiotropy, wherein genetic variants

influence multiple traits through shared heritable factors, presents a particular challenge by potentially mimicking causal relationships. CAUSE differentiates these effects from true causal relationships by utilizing genome-wide summary statistics, thus addressing both types of horizontal pleiotropy and significantly reducing the risk of false-positive results. This approach contrasts with Egger regression and MR-PRESSO, which primarily focus on uncorrelated horizontal pleiotropy by adjusting for pleiotropic effects that are independent of the genetic variant's association with the exposure. CAUSE thereby provides a comprehensive framework for analysis that surpasses traditional MR methods in flexibility and robustness. For model assessment, the expected log pointwise posterior density (ELPD) was used as a Bayesian criterion to evaluate model predictions for new data, with a 'delta elpd' value below zero indicating a preference for the causal model, while a positive value favors the sharing model. DTMR and UVMR analyses were congruent in their assessments of heterogeneity and pleiotropy. Detected outliers were meticulously removed to enhance the precision of our analysis, succeeded by a leave-one-out analysis to ascertain the influence of each SNP on the cumulative findings [39].

To ensure the robustness of the results, a replication dataset was additionally incorporated for further analysis. Subsequently, a meta-analysis of both dataset's outcomes was conducted to establish the final results. Heterogeneity $I^2$ was calculated; if less than 50%, a fixed-effect model was employed, while a random-effects model was used for higher heterogeneity. The mRnd website [40] (https://shiny.cnsgenomics.com/mRnd/) was utilized to evaluate the statistical power of the analyses.

## Results

### Genetic instrument selection

The study conducted MR analyses across two ancestral lineages. In the UVMR analysis, the number of IVs used varied from 35 to 488, explaining genetic variation ranging from 5.83% to 13.92%. Outliers identified by MR-PRESSO were excluded, and the subsequent analyses were conducted with this refined set of IVs. All IVs passed the MR-Steiger filter, satisfying the third hypothesis principle. The average F statistics for the IVs ranged from 121 to 372, demonstrating their robustness, thus reducing the potential bias from weak instruments (**S1 Table**). Scatter plots depicted the direction of causal relationships in the discovery dataset (**Fig 2**), while forest plots described the causal contributions of each IV (**S1 Fig**). In the DTMR analysis,

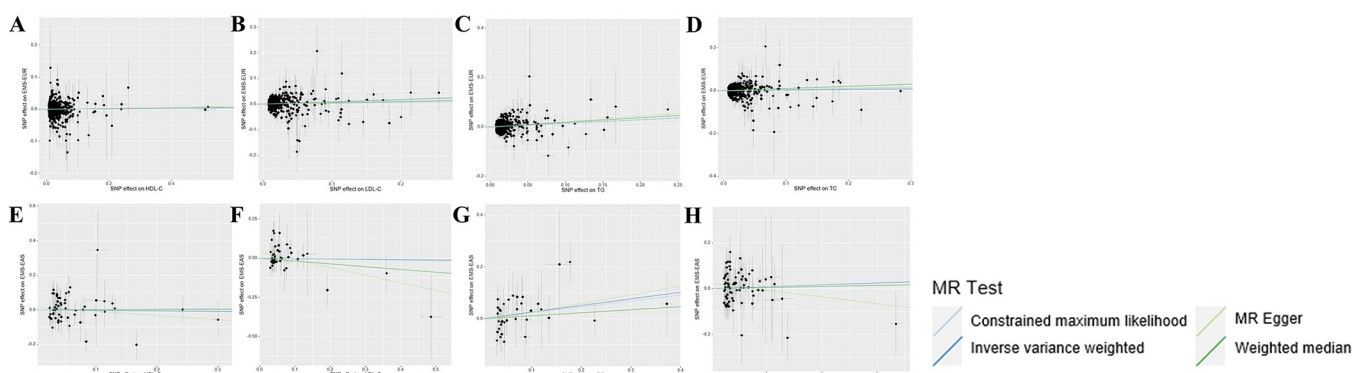

**Fig 2.** Summary of scatterplot results in the discovery dataset (A) HDL-C on EMS-EUR (B) LDL-C on EMS-EUR (C) TG on EMS-EUR (D) TC on EMS-EUR (E) HDL-C on EMS-EAS (F) LDL-C on EMS-EAS (G) TG on EMS-EAS (H) TC on EMS-EAS. LDL-C, Low-Density Lipoprotein Cholesterol; HDL-C, High Density Lipoprotein Cholesterol; TG, Triglyceride; TC, total cholesterol; EMS, endometriosis; EUR, European; EAS, East Asian.

GWAS summary data on TG levels were obtained from the GLGC. A total of 55, 5, 24, and 38 SNPs within or near the genes LPL, PPARA, ANGPTL3, and APOC3, respectively, were selected as the final IVs, with genetic variation explained ranging from 0.04% to 3.99% (S2 Table). The average F statistics for these IVs all exceeded 90.

## Association of genetically predicted exposure with outcome

In the UVMR analysis of EUR ancestry (Fig 3), after FDR correction, the primary IVW method indicated a significant causal association between genetically predicted TG and increased incidence of EMS in the discovery dataset (Odds Ratio [OR] = 1.112, 95%

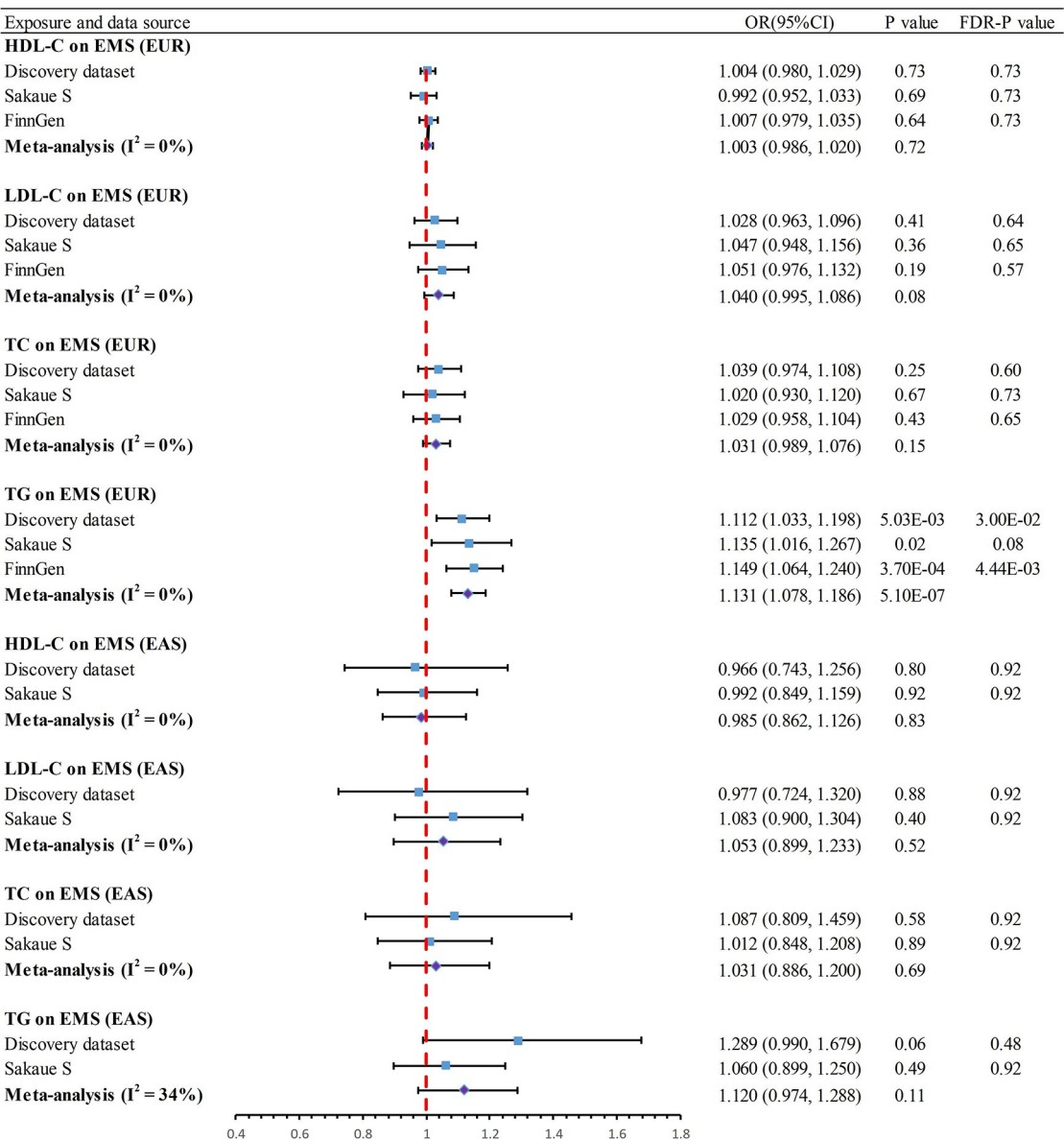

| Exposure and data source | OR(95%CI) | P value | FDR-P value |
|---|---|---|---|
| **HDL-C on EMS (EUR)** | | | |
| Discovery dataset | 1.004 (0.980, 1.029) | 0.73 | 0.73 |
| Sakaue S | 0.992 (0.952, 1.033) | 0.69 | 0.73 |
| FinnGen | 1.007 (0.979, 1.035) | 0.64 | 0.73 |
| **Meta-analysis ($I^2 = 0\%$)** | 1.003 (0.986, 1.020) | 0.72 | |
| **LDL-C on EMS (EUR)** | | | |
| Discovery dataset | 1.028 (0.963, 1.096) | 0.41 | 0.64 |
| Sakaue S | 1.047 (0.948, 1.156) | 0.36 | 0.65 |
| FinnGen | 1.051 (0.976, 1.132) | 0.19 | 0.57 |
| **Meta-analysis ($I^2 = 0\%$)** | 1.040 (0.995, 1.086) | 0.08 | |
| **TC on EMS (EUR)** | | | |
| Discovery dataset | 1.039 (0.974, 1.108) | 0.25 | 0.60 |
| Sakaue S | 1.020 (0.930, 1.120) | 0.67 | 0.73 |
| FinnGen | 1.029 (0.958, 1.104) | 0.43 | 0.65 |
| **Meta-analysis ($I^2 = 0\%$)** | 1.031 (0.989, 1.076) | 0.15 | |
| **TG on EMS (EUR)** | | | |
| Discovery dataset | 1.112 (1.033, 1.198) | 5.03E-03 | 3.00E-02 |
| Sakaue S | 1.135 (1.016, 1.267) | 0.02 | 0.08 |
| FinnGen | 1.149 (1.064, 1.240) | 3.70E-04 | 4.44E-03 |
| **Meta-analysis ($I^2 = 0\%$)** | 1.131 (1.078, 1.186) | 5.10E-07 | |
| **HDL-C on EMS (EAS)** | | | |
| Discovery dataset | 0.966 (0.743, 1.256) | 0.80 | 0.92 |
| Sakaue S | 0.992 (0.849, 1.159) | 0.92 | 0.92 |
| **Meta-analysis ($I^2 = 0\%$)** | 0.985 (0.862, 1.126) | 0.83 | |
| **LDL-C on EMS (EAS)** | | | |
| Discovery dataset | 0.977 (0.724, 1.320) | 0.88 | 0.92 |
| Sakaue S | 1.083 (0.900, 1.304) | 0.40 | 0.92 |
| **Meta-analysis ($I^2 = 0\%$)** | 1.053 (0.899, 1.233) | 0.52 | |
| **TC on EMS (EAS)** | | | |
| Discovery dataset | 1.087 (0.809, 1.459) | 0.58 | 0.92 |
| Sakaue S | 1.012 (0.848, 1.208) | 0.89 | 0.92 |
| **Meta-analysis ($I^2 = 0\%$)** | 1.031 (0.886, 1.200) | 0.69 | |
| **TG on EMS (EAS)** | | | |
| Discovery dataset | 1.289 (0.990, 1.679) | 0.06 | 0.48 |
| Sakaue S | 1.060 (0.899, 1.250) | 0.49 | 0.92 |
| **Meta-analysis ($I^2 = 34\%$)** | 1.120 (0.974, 1.288) | 0.11 | |

**Fig 3. Summary of results of univariable Mendelian randomization.** LDL-C, Low-Density Lipoprotein Cholesterol; HDL-C, High Density Lipoprotein Cholesterol; TG, Triglyceride; FDR, false discovery rate; TC, total cholesterol; EMS, endometriosis; EUR, European; EAS, East Asian.

Confidence Interval [CI] 1.033–1.198, $P = 5.03 \times 10^{-3}$, $P_{FDR} = 0.03$). This finding was consistent with the supplementary CML method (OR = 1.106, 95% CI 1.022–1.197, $P = 0.01$). With an OR of 1.112, there was a 97% statistical power to detect the association between TG and EMS. To evaluate the consistency of the association between EMS and TG across different data sources, the main analysis was replicated based on GWAS studies by Sakaue S et al. (OR = 1.135, 95% CI 1.016–1.267, $P = 0.02$, $P_{FDR} = 0.08$) and the FinnGen consortium (OR = 1.149, 95% CI 1.064–1.240, $P = 3.70 \times 10^{-4}$, $P_{FDR} = 4.44 \times 10^{-3}$). A subsequent meta-analysis (OR = 1.131, 95% CI 1.078–1.186, $P = 5.10 \times 10^{-7}$) of the three datasets further provided consistent causal evidence in line with the discovery dataset findings. However, the primary method did not reveal a causal association for HDL-C (OR = 1.004, 95% CI 0.980–1.029, $P = 0.73$, $P_{FDR} = 0.73$), LDL-C (OR = 1.028, 95% CI 0.963–1.096, $P = 0.41$, $P_{FDR} = 0.64$), or TC (OR = 1.039, 95% CI 0.974–1.108, $P = 0.25$, $P_{FDR} = 0.60$) with EMS. Results from the replication dataset and the meta-analysis were consistent in showing no causal relationship ($P > 0.05$ & $P_{FDR} > 0.05$) (S3 Table).

In the UVMR analysis of EAS ancestry (Fig 3), the IVW method did not detect any causal association effects of HDL-C (OR = 0.966, 95% CI 0.743–1.256, $P = 0.80$, $P_{FDR} = 0.92$), LDL-C (OR = 0.977, 95% CI 0.724–1.320, $P = 0.88$, $P_{FDR} = 0.92$), TC (OR = 1.087, 95% CI 0.809–1.459, $P = 0.58$, $P_{FDR} = 0.92$), or TG (OR = 1.289, 95% CI 0.990–1.679, $P = 0.06$, $P_{FDR} = 0.48$) on EMS. Supplementary methods MR Egger, weighted median and CML also yielded consistent findings (S3 Table). Further replication analyses and meta-analyses provided consistent evidence indicating no causal associations ($P > 0.05$ & $P_{FDR} > 0.05$) (S3 Table).

To ensure the robustness of the results, a series of sensitivity analyses were conducted. The leave-one-out analysis confirmed that the causal relationship was not influenced by any single SNP (S2 Fig), and funnel plots exhibited a symmetric distribution (S3 Fig). MR-Egger analysis indicated no directional pleiotropy in all analyses ($P > 0.05$). MR-PRESSO and Cochran's Q statistics revealed no significant heterogeneity or horizontal pleiotropy in the results ($P > 0.05$), except for LDL-C, TC, and TG in EUR ancestry ($P < 0.05$) (S4 Table).

**Table 2. Adjustment for MVMR analysis in TG on EMS (EUR).**

| Exposure | Outcome | SNP | beta | OR $_{IVW}$ (95%CI) | Pval |
|---|---|---|---|---|---|
| TG | EMS | 491 | 0.108 | 1.114 (1.019, 1.217) | 0.02 |
| SBP | | 491 | 0.006 | 1.006 (1.000, 1.013) | 0.04 |
| TG | EMS | 511 | 0.079 | 1.082 (0.990, 1.183) | 0.08 |
| DBP | | 511 | 0.006 | 1.006 (0.996, 1.016) | 0.23 |
| TG | EMS | 350 | 0.269 | 1.309 (1.139, 1.504) | 1.42E-04 |
| BMI | | 350 | -0.064 | 0.938 (0.845, 1.041) | 0.22 |
| TG | EMS | 167 | 0.153 | 1.165 (1.004, 1.352) | 0.04 |
| IR | | 167 | 0.348 | 1.416 (0.952, 2.107) | 0.08 |
| TG | EMS | 439 | 0.105 | 1.111 (1.020, 1.210) | 0.02 |
| TT | | 439 | -0.132 | 0.877 (0.786, 0.978) | 0.02 |
| TG | EMS | 419 | 0.136 | 1.145 (1.049, 1.251) | 2.49E-03 |
| SHBG | | 419 | 0.046 | 1.047 (0.963, 1.138) | 0.28 |
| TG | EMS | 418 | 0.114 | 1.120 (1.031, 1.217) | 0.01 |
| E2 | | 418 | 0.160 | 1.174 (0.960, 1.435) | 0.12 |

SNP, single nucleotide polymorphism; MVMR, multivariable Mendelian randomization; BMI, body mass index; OR, odds ratio; CI, confidence interval; E2, oestradiol; DPB, diastolic blood pressure; SBP, systolic blood pressure; IR, insulin resistance; TT, total testosterone; SHBG, sex hormone binding globulin; EUR, European; EMS, endometriosis; TG, triglycerides.

**Table 3. Mediation analysis of the mediation effect of TG on EMS via eight confounding factors.**

| Exposure | Mediator | Total effect | Direct effect | Mediation effect | | |
|---|---|---|---|---|---|---|
| | | Effect size (95% CI) | Effect size (95% CI) | Effect size (95% CI) | IE div TE(%) | P |
| TG | SBP | 0.089 (0.011, 0.167) | 0.082 (0.004, 0.161) | 0.007 (0.001, 0.012) | 7.52% | 0.02 |
| | DBP | 0.089 (0.011, 0.167) | 0.087 (0.009, 0.166) | 0.002 (-0.002, 0.006) | 2.01% | 0.36 |
| | BMI | 0.089 (0.011, 0.167) | 0.087 (0.009, 0.165) | 0.002 (-0.002, 0.006) | 2.29% | 0.25 |
| | IR | 0.089 (0.011, 0.167) | 0.098 (0.008, 0.188) | -0.009 (-0.053, 0.035) | -10.20% | 0.67 |
| | TT | 0.089 (0.011, 0.167) | 0.079 (0.001, 0.158) | 0.010 (0.002, 0.017) | 10.79% | 0.01 |
| | SHBG | 0.089 (0.011, 0.167) | 0.087 (0.008, 0.166) | 0.002 (-0.010, 0.013) | 1.85% | 0.77 |
| | E2 | 0.089 (0.011, 0.167) | 0.089 (0.010, 0.167) | 0.000 (-0.004, 0.005) | 0.24% | 0.83 |

EMS, endometriosis; BMI, body mass index; IE div TE, Indirect Effect divided by Total Effect; CI, confidence interval; DPB, Diastolic blood pressure; SBP, Systolic blood pressure; IR, insulin resistance; TT, Total Testosterone; SHBG, Sex hormone binding globulin; E2, Oestradiol; TG, triglycerides.

To further ensure the robustness of the positive findings, a CAUSE analysis was conducted. This analysis indicated that the observed causal effect of TG on EMS in EUR ancestry is robust and not likely to be biased by pleiotropy or heterogeneity (CAUSAL: OR = 1.172, 95% CI 1.096–1.253, $P$ = 0.03, delta elpd = -4.169).

## Mediation & MVMR analysis

In the UVMR analysis, there was evidence of a causal relationship between TG and the risk of EMS in EUR ancestry, achieving statistical significance ($P < 0.05$ & $P_{FDR} < 0.05$). The MVMR analysis (Table 2), considering potential confounding phenotypes, adjusted for SBP, DBP, BMI, IR, TT, SHBG, and E2. After adjusting for DBP, the causal relationship between TG and EMS was no longer significant. This suggests that these confounding factors might partially mediate the causal relationship between TG and EMS. Subsequent mediation MR analysis showed that DBP did not exhibit a mediating effect but acted as a confounder, while SBP (Mediation effect: 7.52%, $P$ = 0.02) and TG (Mediation effect: 10.79%, $P$ = 0.01) were mediators, partially mediating the causal impact of TG on EMS (Table 3). In our analysis, we distinguish between confounders—variables associated with both exposure and outcome but external to their causal pathway—and mediators, which lie within the causal pathway, potentially explaining the observed association between serum lipids and endometriosis. By controlling for confounders and exploring mediators, our study elucidates the direct effects of serum lipids on endometriosis, shedding light on the underlying biological mechanisms.

## Associations of TG-lowering drug targets with EMS

Following positive results in the UVMR analysis, four gene targets for reducing TG were further selected (Fig 4). Specifically, IVW analysis revealed significant causal evidence of a

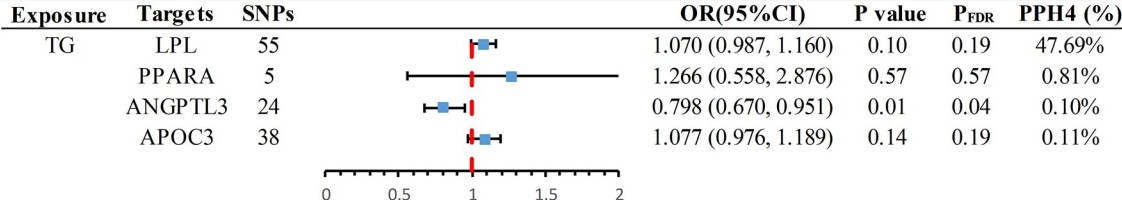

| Exposure | Targets | SNPs | | OR(95%CI) | P value | P_FDR | PPH4 (%) |
|---|---|---|---|---|---|---|---|
| TG | LPL | 55 | | 1.070 (0.987, 1.160) | 0.10 | 0.19 | 47.69% |
| | PPARA | 5 | | 1.266 (0.558, 2.876) | 0.57 | 0.57 | 0.81% |
| | ANGPTL3 | 24 | | 0.798 (0.670, 0.951) | 0.01 | 0.04 | 0.10% |
| | APOC3 | 38 | | 1.077 (0.976, 1.189) | 0.14 | 0.19 | 0.11% |

**Fig 4. MR association between TG mediated by gene PPARA, LPL, ANGPTL3 or APOC3 and EMS(EUR).** MR, Mendelian randomization; PPARA, Peroxisome Proliferator-Activated Receptor Alpha; LPL, lipoprotein lipase; ANGPTL3, angiopoietin-like 3; APOC3, Apolipoprotein C-II; TG, triglycerides; EMS, endometriosis; EUR, European.

relationship between TG mediated by ANGPTL3 (equivalent to an increase of 1 mmol/l) and the risk of EMS onset (OR = 0.798, 95% CI 0.670–0.951, $P$ = 0.01, $P_{FDR}$ = 0.04). Causal evidence of TG mediated by LPL (OR = 1.070, 95% CI 0.987–1.160, $P$ = 0.10, $P_{FDR}$ = 0.19), PPARA (OR = 1.266, 95% CI 0.558–2.876, $P$ = 0.57, $P_{FDR}$ = 0.57), and APOC3 (OR = 1.077, 95% CI 0.976–1.189, $P$ = 0.14, $P_{FDR}$ = 0.19) in relation to EMS was not provided. We conducted a colocalization analysis; however, there was no evidence of colocalization between ANGPTL3 and EMS (PP.H4 = 0.85%), suggesting that the significant MR estimates for these genes may not be robust and should be considered as suggestive evidence only. Sensitivity analyses, including Cochran's Q test, did not reveal any heterogeneity, and MR-Egger regression and MR-PRESSO analysis found no overall horizontal pleiotropy ($P$ > 0.05) (**S5 Table**).

## Discussions

This study conducted a comprehensive multi-ancestry MR analysis to investigate the causal relationships between genetic susceptibility to various serum lipids and the risk of EMS. The MR indicates that higher levels of TG are associated with increased risk of EMS in individuals of EUR ancestry, but this relationship is confounded by DBP and mediated by factors such as SBP and TT. However, the study did not find causal associations between other serum lipids and EMS, a pattern also observed in EAS ancestry. DTMR analysis provided suggestive evidence regarding ANGPTL3 agonists. The significance of these findings will be discussed in terms of inflammatory responses, oxidative stress, mediators, and the comprehensive impact of ANGPTL3 agonists on EMS risk.

Inflammation plays a central role in the pathogenesis of EMS. The research team led by Juan Moreno-Vedia has indicated that high levels of TG in the blood are associated with a chronic inflammatory state, stimulating immune cells such as macrophages to produce pro-inflammatory cytokines [41]. These cytokines not only exacerbate tissue damage but also activate inflammatory pathways, promoting abnormal proliferation and invasiveness of endometrial cells, leading to lesion formation in ectopic endometrial sites. Further evidence from the MR study by Yiting Lin et al. demonstrates a causal relationship between several inflammatory cytokines and an increased incidence of EMS [42]. These inflammatory mediators not only promote local inflammatory responses but can also exacerbate ectopic growth and adhesion of endometrial tissue by affecting hormone levels and cell proliferation. Additionally, Michael Miller and colleagues found that high TG levels might lead to endothelial cell dysfunction [43], which can exacerbate inflammation in the endothelium, increasing leukocyte adhesion and permeation, and thus promoting systemic inflammation. In conditions like obesity and metabolic syndrome, high TG levels are often accompanied by increased inflammation in adipose tissue. Accumulation and activation of inflammatory cells in adipose tissue release pro-inflammatory factors such as prostaglandins and leukotrienes [44]. These inflammatory lipid mediators not only intensify systemic inflammation but may also lead to more severe pain and other related symptoms in EMS patients. Therefore, managing high TG levels can help alleviate inflammatory responses and may reduce the risk of severe symptoms in patients.

In patients with EMS, oxidative stress—an imbalance between free radicals and antioxidants in the body—is considered a significant factor in disease progression [45]. Elevated levels of TG may exacerbate this imbalance, as numerous studies have indicated that excessive lipids can lead to the production of free radicals and oxidative damage. These free radicals can damage cellular membranes, DNA, and proteins, triggering inflammatory pathways within cells [46, 47], and resulting in further damage and functional abnormalities in endometrial tissue. Additionally, research by Luciana Cacciottola et al. suggests that oxidative damage may impair the function of endometrial cells, leading to abnormal cell proliferation and migration,

characteristic features of EMS [48]. Oxidative stress also exacerbates local inflammatory responses by activating inflammation-related transcription factors like NF-kB, promoting the production of inflammatory mediators. These processes collectively contribute to the development and exacerbation of EMS symptoms [49]. Therefore, developing effective strategies to address oxidative stress induced by high triglyceride levels may provide a new therapeutic perspective for alleviating EMS.

In our further mediation MR analysis, we discovered that TT and SBP mediated a portion of the causal effect. Hormonal imbalance is considered a key factor in the pathogenesis of endometriosis, with TT playing a crucial role in regulating the balance between estrogen and progesterone [50]. Research by A. Maclean et al. indicates that changes in testosterone levels can affect the reactivity of the endometrium, encompassing hormone sensitivity as well as cellular proliferation and differentiation, potentially impacting the growth and repair of endometrial tissue [51]. Additionally, studies by Hamed Khalili and colleagues suggest that TT may indirectly influence the development of endometriosis by modulating immune system functionality and oxidative stress responses [52]. Moreover, research by Zhigang Liu emphasizes that fluctuations in TT levels could affect lipid metabolism, including the synthesis and breakdown of triglycerides [53]. SBP, the maximum pressure exerted by blood against vessel walls during heart contraction, can influence the blood supply to pelvic organs, including the uterus [54]. Insufficient blood supply may lead to hypoxia and malnutrition in endometrial tissue, affecting the progression of EMS. Elevated SBP could indicate vascular health issues such as arteriosclerosis or reduced vascular elasticity, potentially causing pelvic microcirculatory disorders [55], adversely affecting the endometrium, and promoting the onset of endometriosis. Therefore, changes in SBP may indirectly affect the growth and survival of ectopic endometrial tissue by influencing the local environment of reproductive organs, particularly the endometrium.

ANGPTL3, a protein primarily expressed in the liver, is commonly targeted by inhibitors as an effective strategy for treating hypertriglyceridemia and other lipid metabolism disorders [56]. However, our DTMR study presents a different perspective, indicating that activation of ANGPTL3 (via ANGPTL3 agonists) may be associated with reduced risk of EMS, as evidenced by suggestive findings. A DiOGenes study by Anne Lundby Hess points to potential roles of ANGPTL3 beyond lipid metabolism, such as influencing hormone levels, and regulating immune responses and inflammatory processes [57]. Additionally, ANGPTL3 might influence the risk of EMS through mechanisms not yet fully understood, including effects on cellular proliferation, apoptosis, and the migration and adhesion of endometrial cells. These findings highlight the possible unknown biological mechanisms of ANGPTL3, suggesting the need for more in-depth biological research and data analysis to fully understand how ANGPTL3 and changes in TG may impact EMS risk in unexpected ways.

This study possesses several distinct advantages. First, it represents the inaugural application of MR analysis to establish the causal relationship between serum lipid levels and the risk of EMS in both EUR and EAS ancestries. Second, our analysis employed robust methodologies, with all F statistics exceeding 10, thereby mitigating potential biases due to weak instruments. Third, the robustness of the primary findings was evaluated using various statistical models in sensitivity analyses. The CAUSE analysis was implemented to avoid biases caused by heterogeneity and pleiotropy, enhancing the reliability of the evidence. Additionally, replication datasets and meta-analyses corroborated the main findings. Fourth, the study enriched its framework by incorporating multiple MR analyses. However, our research is not without limitations. A significant constraint is the inability to eliminate horizontal pleiotropy through MR-PRESSO analysis; nevertheless, additional sensitivity assessments confirmed the robustness of the association evidence. Another limitation is that the study only accessed aggregate

data, precluding subgroup analyses or stratification by age and gender. This is particularly relevant as our analysis could not differentiate the potentially unique causal relationships in males and females, nor could it account for the variations in disease risk and lipid levels across different ages. Finally, due to limited GWAS sample data in EAS ancestry, our analysis might not have had sufficient power to detect more subtle causal relationships.

## Conclusions

In summary, this MR study established a causal relationship between genetically predicted levels of TG and increased risk of EMS in EUR ancestry. It emphasizes the need for multifaceted prevention strategies. Further research is required to explore the potential mechanisms of ANGPTL3's role in EMS.

## Supporting information

**S1 Table. Power calculations for univariable Mendelian randomization analyses (Discovery dataset).**
(XLSX)

**S2 Table. Information of genetic instrumental variants associated with TG located within 100 kb windows from gene LPL, PPARA, ANGPTL3 or APOC3.**
(XLSX)

**S3 Table. Summary of MR analysis results.**
(XLSX)

**S4 Table. Summary of sensitivity results.**
(XLSX)

**S5 Table. MR association between TG mediated by gene PPARA, LPL, ANGPTL3 or APOC3 and EMS(EUR).**
(XLSX)

**S1 Fig. Summary of forest map results.** (A) HDL-C on EMS-EUR (B) LDL-C on EMS-EUR (C) TG on EMS-EUR (D) TC on EMS-EUR (E) HDL-C on EMS-EAS (F) LDL-C on EMS-EAS (G) TG on EMS-EAS (H) TC on EMS-EAS. LDL-C, Low-Density Lipoprotein Cholesterol; HDL-C, High Density Lipoprotein Cholesterol; TG, Triglyceride; TC, total cholesterol; EMS, endometriosis; EUR, European; EAS, East Asian.
(TIF)

**S2 Fig. Summary of results of the leave-one-out method.** (A) HDL-C on EMS-EUR (B) LDL-C on EMS-EUR (C) TG on EMS-EUR (D) TC on EMS-EUR (E) HDL-C on EMS-EAS (F) LDL-C on EMS-EAS (G) TG on EMS-EAS (H) TC on EMS-EAS. LDL-C, Low-Density Lipoprotein Cholesterol; HDL-C, High Density Lipoprotein Cholesterol; TG, Triglyceride; TC, total cholesterol; EMS, endometriosis; EUR, European; EAS, East Asian.
(TIF)

**S3 Fig. Summary of funnel plot results.** (A) HDL-C on EMS-EUR (B) LDL-C on EMS-EUR (C) TG on EMS-EUR (D) TC on EMS-EUR (E) HDL-C on EMS-EAS (F) LDL-C on EMS-EAS (G) TG on EMS-EAS (H) TC on EMS-EAS. LDL-C, Low-Density Lipoprotein Cholesterol; HDL-C, High Density Lipoprotein Cholesterol; TG, Triglyceride; TC, total cholesterol; EMS, endometriosis; EUR, European; EAS, East Asian.
(TIF)

## Acknowledgments

All data are freely available from the literature cited in the article, or publicly from Table 1. In the EUR pedigree, HDL-C (GCST90239652),LDL-C (GCST90239658),TC (GCST90239676), TG (GCST90239664) were expressed as HDL-C (GCST90239651),LDL-C (GCST90239657), TC in the EAS pedigree (GCST90239675) and TG (GCST90239663) are all available for download at https://www.ebi.ac.uk/gwas/ based on their IDs. The EMS data of Rahmioglu N et al. in the discovery dataset are available for download at the above website under GCST90269970. The EMS of BBJ is available for download at https:// under the code "bbj-a-114". gwas.mrcieu. ac.uk/. In the replication dataset, the EMS of the FinnGen consortium can be downloaded at https://www.finngen.fi/en. The EMS of Sakaue S et al. can be identified and downloaded under the codes "ebi-a-GCST90018839" and " ebi-a-GCST90018619" respectively, which can be identified and downloaded at https://gwas.mrcieu.ac.uk/. SBP(ieu-b-38),DBP(ieu-b-39), BMI(ieu-b-40),IR(ieu-b-118),TT(ieu-b-4864),SHBG (ieu-b-4870),E2(ieu-b-4872) can all be recognized and downloaded from https://gwas.mrcieu.ac.uk/ according to the relevant codes. The authors would like to thank all researchers for sharing these data.

## Author Contributions

**Conceptualization:** Hongling Zhang.

**Data curation:** Hongling Zhang, Yawei Fan.

**Formal analysis:** Hongling Zhang.

**Investigation:** Hongling Zhang, Daoyuan Yue.

**Methodology:** Hongling Zhang, Xiaoqing Feng.

**Resources:** Daoyuan Yue.

**Software:** Huijun Li.

**Supervision:** Huijun Li, Daoyuan Yue.

**Validation:** Huijun Li.

**Visualization:** Yawei Fan.

**Writing – original draft:** Hongling Zhang.

**Writing – review & editing:** Hongling Zhang, Yawei Fan.

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
