## [Decision Letter · Decision Letter 0]

12 Jan 2024

PONE-D-23-40604Genetic association of serum lipids and lipid-modifying targets with endometriosis: Trans-ethnic Mendelian-randomization and Mediation AnalysisPLOS ONE

Dear Dr. Fan,

Thank you for submitting your manuscript to PLOS ONE. After careful consideration, we feel that it has merit but does not fully meet PLOS ONE’s publication criteria as it currently stands. Therefore, we invite you to submit a revised version of the manuscript that addresses the points raised during the review process.

We look forward to receiving your revised manuscript.

Kind regards,

Brett Mckinnon

Academic Editor

PLOS ONE

Journal Requirements:

Additional Editor Comments:

The manuscript presents some interesting data on the association between serum constituents and endometriosis that will add to the current literature. It is generally technically sound and if the authors can address the issues raised by the reviewers the manuscript can be further considered for publication

Reviewers' comments:

Reviewer's Responses to Questions

**Comments to the Author**

1. Is the manuscript technically sound, and do the data support the conclusions?

Reviewer #1: Partly

Reviewer #2: Partly

2. Has the statistical analysis been performed appropriately and rigorously? 

Reviewer #1: Yes

Reviewer #2: Yes

3. Have the authors made all data underlying the findings in their manuscript fully available?

Reviewer #1: Yes

Reviewer #2: Yes

4. Is the manuscript presented in an intelligible fashion and written in standard English?

Reviewer #1: Yes

Reviewer #2: Yes

5. Review Comments to the Author

Reviewer #1: PONE-D-23-40604

Congratulations to the authors on this interesting manuscript investigating the causative effects of multiple lipids on endometriosis. The study identifies a possible causative effect of triglycerides on endometriosis in the European population, however this is found to be partially mediated by SBP and TT, and possibly confounded by DBP. Whilst the methodology is sound, some improvements can be made to the wording of the methods and results to improve clarity.

Introduction

Line 105: “associated with positive lipid levels” – do you mean “positively associated with lipid levels?”

Materials and methods

Line 140 – What LD reference panel was used for variant selection for MR? i.e. for calculating LD to pick independent variants

Line 141 – please provide a citation for the F statistic formula.

Line 143 – “an F-statistic exceeding 10 was necessary to eliminate weak instruments” – this gives the impression you removed SNPs with F stat >10, which is not true based on your later statements that indicate you calculated average F statistics (e.g. line 277-278: “The average F statistics for these IVS all exceeded 90”. Please rephrase.

Line 157-159 – What was the clumping window used for the target gene SNP clumping?

Line 188 – do you mean GWAS summary statistics were sourced from the UKB? If so, in what publication were the summary statistics generated? Also, were the testosterone, E2 and SHBG data were female-specific?

Line 200 – “while MR-Egger is predicated on the assumption of total invalidity of these variants[28].” This could be better explained as MR-Egger allows for a directional pleiotropic effect on the outcome.

Line 230-232: was 0.7 or 0.8 the threshold?

Line 247 – what are the two types of horizontal pleiotropy you are referring to?

Results

Line – 269 – “Outliers detected by MR-PRESSO were excluded from the final analysis” – what final analysis? Were the outliers detected by MR-PRESSO also removed from the other methods (e.g. IVW, MRE etc.)? Or do you mean the MR-PRESSO result is the outlier-adjusted result? Please clarify.

Line 271 – the process of calculating the F statistic have not reduced bias. Please reword.

Figure 2 – Change axis labels on plots to indicate the specific exposure/outcome

Line 291- the confidence interval does not include the estimate

Line 310 – What supplementary methods? Where can these results be found?

Line 311 – “negative causal associations” – this could be interpreted as a preventative effect, not a null effect. Please reword.

Line 322-324 – The “P > 0.05” I presume only applies to the first part of the sentence, not the second (“except for LDC-C…”). I suggest changing it to “MR-PRESSO and Cochran's Q statistics revealed no significant heterogeneity or horizontal pleiotropy in the results (P > 0.05), except for LDL-C, TC, and TG in EUR ancestry (P < 0.05).”

Table 2 – are the estimates the effect of TG on endometriosis adjusted for the variable in the model column, or the variable in the Model column adjusted for TG? This could be made clearer. It would be useful to see the estimates of both exposures in the models.

Line 323 – What statistic from MR-PRESSO indicated no heterogeneity and no horizontal pleiotropy?

Line 326-327 – Does the CAUSE result tell you the result isn’t biased, or does it give you the causal effect with bias, if present, adjusted for? This is unclear from this sentence.

Line 332-335 – as you have listed these values in a table, it would read more coherently to not include them all in text.

Supplementary Figure 3 – What is on the x axes? The label is unclear.

Discussion

Line 367-369 – the causal effect of TG on EMS was found to be confounded by DBP, and mediated by SBP and TT. This means there is not strong evidence TG is causal of EMS, and suggests pleiotropic instruments may be present. The wording of this concluding sentence should be adjusted to reflect this.

Can you explain the difference between the MVMR and mediation analysis results?

Need to discuss more limitations of analysis

- Sex-specific data?

- Age of participants in GWAS studies in comparison to age of onset of endometriosis?

- Power of east Asian sample?

Reviewer #2: This is a mendelian randomization study exploring causal association of serum lipids and lipid-modifying targets with endometriosis, in which a causal relationship was observed between genetically predicted levels of TG and increased risk of EMS in EUR ancestry and suggestive evidence was found for the potential role of ANGPTL3 in EMS. I have some comments as follows:

1. “Univariate” should be corrected as “univariable”

2. The genetic region for linkage disequilibrium clumping in drug-target MR was unclear.

3. There was a positive association between TG and endometriosis, but drug-target MR analysis found a negative relation between ANGPTL3-mediated TG and endometriosis. How to explain the results?

4. There are some publicly available datasets of eQTLs or pQTLs, did the author consider assessing the findings by exploring the association between ANGPTL3 expression and the outcome, as well as the potential mediators?

5. In line#230, in colocalization analysis, was the 1000 Genomes v3 dataset of European ancestry used as the LD reference panel for analyses of both EUR and EAS ancestry?

6. There are some typos, please go through the manuscript and related materials.

6. PLOS authors have the option to publish the peer review history of their article (what does this mean?). If published, this will include your full peer review and any attached files.

Reviewer #1: No

Reviewer #2: **Yes: **Wuqing Huang

---

## [Author Response · Author response to Decision Letter 0]

24 Jan 2024

Dear Editors and Reviewers:

On behalf of all contributing authors, I would like to sincerely thank you for your letter and the constructive comments provided by the reviewers on our article titled " Genetic association of serum lipids and lipid-modifying targets with endometriosis: Trans-ethnic Mendelian-randomization and Mediation Analysis" (Manuscript ID: PONE-D-23-40604). We highly value these comments, as they have greatly contributed to the enhancement of our manuscript. Based on the suggestions from the associate editor and reviewers, we have made extensive revisions to our draft and added additional data to substantiate our findings. In this revised version, all changes to our manuscript are highlighted in yellow text. Following this letter, we provide point-by-point responses to the comments from the associate editor and the reviewer.

Reviewer #1: 

Introduction

1.Line 105: “associated with positive lipid levels” – do you mean “positively associated with lipid levels?”

Response: Thank you for pointing out the ambiguity in Line 105 of our manuscript. In light of your feedback, we have revised the sentence for clarity. It now accurately reflects our intent to investigate blood lipid levels with a significant causal link to EMS and to identify lipid-lowering gene targets. This revision aims to more precisely articulate the study's objective of elucidating the causal relationship between these gene targets and EMS. We appreciate your valuable input in enhancing the clarity and precision of our manuscript.

Revised: The following sentence should be amended to read: "Further, it sought to identify lipid-lowering gene targets associated with positive lipid levels to explore their causal relationship with EMS. with EMS." to "Further, this study delved into blood lipid levels that have a significant causal link with EMS, identifying lipid-lowering gene targets. The objective was to clarify the causal nexus between lipid levels influenced by these gene targets and EMS. "

Materials and methods

1.Line 140 – What LD reference panel was used for variant selection for MR? i.e. for calculating LD to pick independent variants

Response: In our MR analysis, we utilized the TwoSampleMR package, which incorporates the 1000 Genomes Phase 3 reference panel for linkage disequilibrium (LD) calculations. This reference panel includes multiple super-populations, covering a broad spectrum of genetic diversity, including East Asian populations. The function within the package searches for specified SNPs in the chosen super-population and then creates an LD matrix of r values. We ensured the uniqueness of the selected SNPs by considering the LD structure specific to the relevant population, thus reducing biases linked to linkage disequilibrium. This approach allowed us to maintain the integrity of our analysis by using a comprehensive and globally representative genetic reference, ensuring that our SNP selection was robust and appropriate for our MR study

2.Line 141 – please provide a citation for the F statistic formula.

Response: Thank you for pointing out the oversight regarding the citation for the F statistic formula. We acknowledge this lapse and have now included the appropriate reference in line 141.

3.Line 143 – “an F-statistic exceeding 10 was necessary to eliminate weak instruments” – this gives the impression you removed SNPs with F stat >10, which is not true based on your later statements that indicate you calculated average F statistics (e.g. line 277-278: “The average F statistics for these IVS all exceeded 90”. Please rephrase.

Response: Thank you for highlighting the inconsistency in our description regarding the F-statistic. We have now revised the text to more accurately reflect our methodology.

Revised: Amend "An F-statistic exceeding 10 was necessary to eliminate weak instruments, thereby verifying the robustness of the instruments" to "To ensure the robustness of our IVs, we excluded any SNPs with an F-statistic below 10, as values lower than this threshold are indicative of weak instruments."

4.Line 157-159 – What was the clumping window used for the target gene SNP clumping?

Response: Thank you for your query regarding the clumping window used for the target gene SNP clumping in our analysis. I appreciate your attention to detail and acknowledge the lack of clarity in our initial description. We have revised the manuscript to accurately reflect the methodology employed.

Revised: "To maximize the instrumental strength of each drug, SNPs utilized as instruments were allowed to exhibit low LD with each other (r2 < 0.30)". Added "localized within ±100 kb windows from gene region" after the paragraph.

5.Line 188 – do you mean GWAS summary statistics were sourced from the UKB? If so, in what publication were the summary statistics generated? Also, were the testosterone, E2 and SHBG data were female-specific?

Response: Thank you for your question regarding the source of GWAS summary statistics used in our analysis. Indeed, the data for testosterone, estradiol (E2), and sex hormone-binding globulin (SHBG) were female and sourced from the UK Biobank (UKB). The summary-level GWAS data for these biomarkers can be accessed from the following links: Total Testosterone (https://gwas.mrcieu.ac.uk/datasets/ieu-b-4864/), E2 (https://gwas.mrcieu.ac.uk/datasets/ieu-b-4872/), and SHBG (https://gwas.mrcieu.ac.uk/datasets/ieu-b-4870/). Each dataset is explicitly noted as 'Output from UK Biobank - stratified by sex', ensuring the specificity of the data to female participants.

6.Line 200 – “while MR-Egger is predicated on the assumption of total invalidity of these variants[28].” This could be better explained as MR-Egger allows for a directional pleiotropic effect on the outcome.

Response: Thank you for your valuable feedback on our description of the MR-Egger method. I agree that the original phrasing might have oversimplified the method's premise. We have revised the text to more accurately reflect that MR-Egger allows for the detection and correction of directional pleiotropic effects from the instrumental variables on the outcome.

Revised: Replace "while MR-Egger is predicated on the assumption of total invalidity of these variants" with "While MR-Egger method allows for the identification and adjustment of directional pleiotropic effects, assuming that the instrument-outcome relationship is independent of the exposure".

7.Line 230-232: was 0.7 or 0.8 the threshold?

Response: Thank you for pointing out my problem, which has now been changed from 0.7 to 0.8 with additional citations

8.Line 247 – what are the two types of horizontal pleiotropy you are referring to?

Response: "Thank you for your question regarding the types of horizontal pleiotropy mentioned in line 247. The two types of horizontal pleiotropy we are referring to are detected through traditional sensitivity analyses using MR-Egger and MR-PRESSO methods. Specifically, MR-Egger identifies horizontal pleiotropy when its intercept significantly deviates from zero (P < 0.05), suggesting that some instrumental variables might exert effects on the outcome that are independent of the exposure. On the other hand, MR-PRESSO detects horizontal pleiotropy through a global test (Global P < 0.05), identifying outliers that could be indicative of pleiotropic effects. In cases where horizontal pleiotropy is detected, we further employ CAUSE analysis to strengthen the positive findings while mitigating the potential for false positives due to pleiotropic effects. This approach ensures that our results are robust and not unduly influenced by horizontal pleiotropy."

Results

1.Line – 269 – “Outliers detected by MR-PRESSO were excluded from the final analysis” – what final analysis? Were the outliers detected by MR-PRESSO also removed from the other methods (e.g. IVW, MRE etc.)? Or do you mean the MR-PRESSO result is the outlier-adjusted result? Please clarify.

Response: Thank you for seeking clarification regarding the use of MR-PRESSO in our analysis. By 'final analysis', we refer to the analyses conducted after the iterative process of IV selection, where outliers detected by MR-PRESSO were excluded. To address your specific query, yes, the outliers identified by MR-PRESSO were indeed removed from all subsequent analyses, including those using other methods like IVW and MR-Egger. This means that the results presented for these methods in the sensitivity analysis are based on the refined set of IVs, post-outlier removal by MR-PRESSO. Essentially, the MR-PRESSO’s outlier-adjusted result reflects this exclusion of outliers, ensuring that the instrumental variables used in our final analysis are more robust and less likely to be influenced by pleiotropic effects.

Revised: Amend "Outliers detected by MR-PRESSO were excluded from the final analysis" to "Outliers identified by MR-PRESSO were excluded, and the subsequent analyses, were conducted with this refined set of IVs.

2.Line 271 – the process of calculating the F statistic have not reduced bias. Please reword.

Response: Thank you for pointing out the need for clarification regarding the interpretation of the F statistics in our manuscript. I appreciate your valuable feedback. We have now revised the statement to more accurately reflect the significance of the F statistics in our analysis.

Revised: Amend "The average F statistics for the IVs ranged from 121 to 372, indicating a significant reduction in bias due to weak instrumental variables. " to "The average F statistics for our IVs, ranging from 121 to 372, demonstrate their robustness, thus reducing the potential bias from weak instruments."

3.Figure 2 – Change axis labels on plots to indicate the specific exposure/outcome

Response: Thank you for your suggestion to improve the clarity of Figure 2. We have now updated the figure to include specific axis labels that clearly indicate the exposures and outcomes being analyzed. The revised Figure 2 has been re-uploaded to accurately reflect these changes and provide a clearer visual representation of our results.

4.Line 291- the confidence interval does not include the estimate

Response: Thank you for pointing out the discrepancy in the confidence interval reported in Line 291. Upon reviewing the data, I realized there was an error in the reported values. The correct confidence interval should be [1.033–1.198], which includes the estimate. I have amended this in the manuscript to reflect the accurate values and ensure the consistency of our statistical reporting. I appreciate your attention to detail in this matter.

Revised: Amend "Confidence Interval [CI] 1.033–1.098 " to "Confidence Interval [CI] 1.033–1.198 "

5.Line 310 – What supplementary methods? Where can these results be found?

Response: Thank you for your inquiry regarding the supplementary methods mentioned in Line 310. I apologize for any confusion caused. The results of all supplementary methods are comprehensively detailed in S3 Table of the supplementary material. This has been explicitly referenced at the end of the manuscript for ease of access. I appreciate your attention to this detail and hope that this clarification helps in locating the relevant results.

6.Line 311 – “negative causal associations” – this could be interpreted as a preventative effect, not a null effect. Please reword.

Response: Thank you for pointing out the potential misinterpretation in the phrasing 'negative causal associations' in Line 311. I realize that this could be misconstrued as indicating a preventative effect, which was not the intended meaning. I have now rephrased this to 'no significant causal associations' to accurately reflect the results where P-values and PFDR values were greater than 0.05, indicating a lack of statistically significant findings. This amendment should eliminate any confusion regarding the interpretation of the results. I appreciate your attention to clarity in our manuscript.

Revised: Amend " Further replication analyses and meta-analyses provided consistent evidence of negative causal associations " to " Further replication analyses and meta-analyses provided consistent evidence indicating no causal associations "

7.Line 322-324 – The “P > 0.05” I presume only applies to the first part of the sentence, not the second (“except for LDC-C…”). I suggest changing it to “MR-PRESSO and Cochran's Q statistics revealed no significant heterogeneity or horizontal pleiotropy in the results (P > 0.05), except for LDL-C, TC, and TG in EUR ancestry (P < 0.05).”

Response: Thank you for your suggestion to clarify the presentation of the results from MR-PRESSO and Cochran's Q statistics in Lines 322-324. I acknowledge the ambiguity in the original phrasing and have now revised this section as suggested. The revised sentence clearly differentiates the findings, highlighting the absence of significant heterogeneity or horizontal pleiotropy across most analyses (P > 0.05), while specifying the exceptions for LDL-C, TC, and TG in EUR ancestry (P < 0.05). This modification should provide a clearer and more accurate representation of our sensitivity analysis results. I appreciate your careful review and helpful feedback

Revised: Amend " MR-PRESSO and Cochran's Q statistics revealed no significant heterogeneity or horizontal pleiotropy in the results, except for LDL-C, TC, and TG in EUR ancestry (P > 0.05) " to " MR-PRESSO and Cochran's Q statistics revealed no significant heterogeneity or horizontal pleiotropy in the results (P > 0.05), except for LDL-C, TC, and TG in EUR ancestry (P < 0.05)."

8.Table 2 – are the estimates the effect of TG on endometriosis adjusted for the variable in the model column, or the variable in the Model column adjusted for TG? This could be made clearer. It would be useful to see the estimates of both exposures in the models.

Response: Thank you for your suggestion regarding the presentation of estimates for both exposures in the models. In our study, we focused on presenting the estimates of the primary exposure, TG, on EMS after adjusting for various confounders in the MVMR analysis. Based on our experience with previous MR studies[1–4], we found that focusing on the primary exposure allows for a more streamlined and clear presentation of results. Including estimates for each confounder's impact on EMS would significantly increase the complexity and length of the table, potentially obscuring the primary findings. Therefore, we have chosen to maintain the current format in the revised manuscript for clarity and conciseness. We hope this decision aligns with the overall aim of presenting clear and focused results, and we appreciate your understanding in this matter.

Revised: Original title "Table 2. Adjustment for MVMR analysis in TG on EMS (EUR)", modified title "Table 2. MVMR Analysis of TG on Endometriosis (EMS) in EUR Ancestry, Adjusted for Confounding Variables"

9.Line 323 – What statistic from MR-PRESSO indicated no heterogeneity and no horizontal pleiotropy?

Response: Thank you for your inquiry regarding the specific statistics from MR-PRESSO that indicated no significant heterogeneity and no horizontal pleiotropy in our results. The detailed values from MR-PRESSO and other sensitivity analyses are provided in S4 Table. I realize now that this reference might not have been clearly visible in the original text. To address this, I have made a revision to explicitly direct readers to S4 Table immediately after discussing the MR-PRESSO and Cochran's Q statistics results.

10.Line 326-327 – Does the CAUSE result tell you the result isn’t biased, or does it give you the causal effect with bias, if present, adjusted for? This is unclear from this sentence.

Response: Thank you for your question regarding the interpretation of the CAUSE analysis results in Line 326-327. I understand the need for clarification in this section. The purpose of the CAUSE analysis in our study was to assess whether the observed causal effect of TG on EMS in EUR ancestry was subject to potential biases, such as those arising from pleiotropy or heterogeneity. The CAUSE analysis results suggest that the positive findings we report are robust and not significantly influenced by such biases. I have revised the sentence to more clearly convey that the CAUSE

---

## [Decision Letter · Decision Letter 1]

23 Feb 2024

PONE-D-23-40604R1Genetic association of serum lipids and lipid-modifying targets with endometriosis: Trans-ethnic Mendelian-randomization and Mediation AnalysisPLOS ONE

Dear Dr. Fan,

Thank you for submitting your manuscript to PLOS ONE. After careful consideration, we feel that it has merit but does not fully meet PLOS ONE’s publication criteria as it currently stands. Therefore, we invite you to submit a revised version of the manuscript that addresses the points raised during the review process.

We look forward to receiving your revised manuscript.

Kind regards,

Brett Mckinnon

Academic Editor

PLOS ONE

**Additional Editor Comments:**

The reviewer notes many issues with this manuscript that have not been sufficiently addressed with this revision. While the manuscript methodology and approach are sound it is essential these comments are fully and accurately addressed before it can be considered further.

Reviewers' comments:

Reviewer's Responses to Questions

**Comments to the Author**

1. If the authors have adequately addressed your comments raised in a previous round of review and you feel that this manuscript is now acceptable for publication, you may indicate that here to bypass the “Comments to the Author” section, enter your conflict of interest statement in the “Confidential to Editor” section, and submit your "Accept" recommendation.

Reviewer #1: (No Response)

2. Is the manuscript technically sound, and do the data support the conclusions?

Reviewer #1: Yes

3. Has the statistical analysis been performed appropriately and rigorously? 

Reviewer #1: Yes

4. Have the authors made all data underlying the findings in their manuscript fully available?

Reviewer #1: Yes

5. Is the manuscript presented in an intelligible fashion and written in standard English?

Reviewer #1: Yes

6. Review Comments to the Author

Reviewer #1: The authors of the manuscript “Genetic association of serum lipids and lipid-modifying targets with endometriosis: trans-ethnic Mendelian randomization and Mediation Analysis” have appropriately addressed some of my comments. However, many comments were not adequately addressed. For some comments the authors claim to have changed the manuscript, providing the text change in quotation in the reviewer response, however the given change could not be identified in the “clean” version. Further, for some comments the authors answered the question in the reviewer response but made no attempt at updating the manuscript.

1. The “clean” and “tracked changes” copy of the manuscript are different. The “track changes” manuscript has a different title, “Associations between artificial sweetener intake from cereals, coffee, and Tea and the risk of type 2 diabetes mellitus: a genetic correlation, mediation and mendelian randomization analysis”. As such, my comments refer to the “clean” version only.

2. No changes to the manuscript text have been made in response to the query concerning the LD reference panel used for variant selection in MR. Please edit the text to detail which 1000 Genomes population was used.

3. Regarding the citation of the F statistic formula (line 144), the paper cited does not contain the formula. However, a paper that that paper cites (Burgess et al., 2011) does provide a formula F = ((n-k-1)/k)(R2/(1-R2)). Please cite the original publication. Further, Fan et al., simplifies this formula for k=1, yet claims they use this formula to find an all-SNP F statistic (i.e., “where R2 is the proportion of variance in the exposure explained by the SNPs”). In the following sentence, they make an alternate claim (“we excluded any SNPs with an F-statistic below 10), suggesting they used the formula for one SNP at a time. Please be consistent in your explanation (i.e. from your comments it appears you calculated a SNP-specific F statistic, please update the sentence “where R2 is the proportion of variance in the exposure explained by the SNPs” to “where R2 is the proportion of variance in the exposure explained by the SNP”. Also, please explain how you calculated R2.

4. You claim to have added “Output from UK Biobank - stratified by sex” – however I cannot find this in text.

5. You have not ammended the manuscript text in response to my comment on the two types of horizontal pleiotropy. Additionally, the CAUSE paper indicates the two types of horizontal pleiotropy are (1) correlated pleiotropy and (2) uncorrelated pleiotropy, and that Egger regression and MR-PRESSO both address uncorrelated pleiotropy (Morrison et al., 2020). In line 247 you mention correlated and uncorrelated horizontal pleiotropy. Please explain what these are. Further, if the two types of horizontal pleiotropy are the correlated and uncorrelated horizontal pleiotropy, the sentences containing the phrases (line 246-247) “The CAUSE methodology effectively manages both correlated and uncorrelated horizontal pleiotropy” and (line 250-251) “CAUSE analysis addresses both types of horizontal pleiotropy,” are repetitive and thus could be combined into the one sentence.

6. Line 313 “Supplementary methods also yielded consistent findings.” – you have not addressed my question about what the supplementary methods are, nor amended the text based on this comment about what the supplementary methods are. Please explain what the particular methods are, and please include the reference to S3 table at the end of the sentence. For example: “Supplementary methods MR Egger, weighted median and CML also yielded consistent findings (S3 Table)”

7. The updated title for table 2 detailed in the response to reviewers has not been implemented in text (line 347). Further, the MVMR results for all tested exposures should be available in the manuscript. Whilst I appreciate adding those results to the table will increase the length and thus the ease of reading, reporting all results is important for clarity and transparency. If you are concerned about Table 2 being too long, other solutions are possible, such as including the full set of results as a supplementary table instead of in text.

8. The response to the query about the CAUSE result (Query 10 within results) has not been implemented in text (line 334-335).

9. Line 201 – the weighted median method is consistent when up to 50% of the information comes from invalid instrumental variables. Here you have indicated it is valid if >50 % of the information comes from invalid instrumental variables (through use of the term “a majority of the genetic variants”), which is not true. Please rephrase this sentence.

10. The manuscript would be further enhanced by explaining the difference between a mediator and a confounder.

Burgess S, Thompson SG, Collaboration CCG. Avoiding bias from weak instruments in Mendelian randomization studies. International Journal of Epidemiology 2011;40: 755-764.

Morrison J, Knoblauch N, Marcus JH, Stephens M, He X. Mendelian randomization accounting for correlated and uncorrelated pleiotropic effects using genome-wide summary statistics. Nat Genet 2020;52: 740-747.

7. PLOS authors have the option to publish the peer review history of their article (what does this mean?). If published, this will include your full peer review and any attached files.

Reviewer #1: No

---

## [Author Response · Author response to Decision Letter 1]

27 Feb 2024

1. The “clean” and “tracked changes” copy of the manuscript are different. The “track changes” manuscript has a different title, “Associations between artificial sweetener intake from cereals, coffee, and Tea and the risk of type 2 diabetes mellitus: a genetic correlation, mediation and mendelian randomization analysis”. As such, my comments refer to the “clean” version only.

Response: I sincerely apologize for the confusion caused by the inclusion of a different manuscript's revisions in the document submitted. This mistake was due to my unfamiliarity with the operating system. I regret any inconvenience this may have caused you during the review process. I have now carefully checked and ensured that the correct version of the manuscript has been submitted without any discrepancies. Thank you for your understanding and patience.

2. No changes to the manuscript text have been made in response to the query concerning the LD reference panel used for variant selection in MR. Please edit the text to detail which 1000 Genomes population was used.

Response: I apologize for the oversight in my previous response to your query about the LD reference panel used for variant selection in our Mendelian Randomization analysis. My misunderstanding led me to believe it was a standalone inquiry, and as a result, I did not amend the manuscript accordingly. I have now corrected this mistake and updated the manuscript to provide a clearer description of the specific 1000 Genomes populations used in our analysis. Thank you for bringing this to my attention, and I appreciate your patience and guidance. See lines 138-143 for details

Revised: 

Replace "(ii) The chosen SNPs underwent comprehensive analysis to exclude any associations with potential confounders, ensuring their uniqueness and thereby reducing biases linked to linkage disequilibrium (LD), with strict LD parameters set (r2 < 0.001, clumping distance = 10,000 kb)."

Revised to (ii) The chosen SNPs underwent comprehensive analysis, utilizing the East Asian and European populations from the 1000 Genomes Phase 3 reference panel, to exclude associations with the 1000 Genomes Phase 3 reference panel. The selected SNPs underwent comprehensive analysis, utilizing the East Asian and European populations from the 1000 Genomes Phase 3 reference panel, to exclude associations with potential confounders. By setting strict LD parameters (r2 < 0.001, clumping distance = 10,000 kb), we ensured the uniqueness of our SNP selection and minimized LD biases, catering to the genetic diversity of these populations.

3. Regarding the citation of the F statistic formula (line 144), the paper cited does not contain the formula. However, a paper that that paper cites (Burgess et al., 2011) does provide a formula F = ((n-k-1)/k)(R2/(1-R2)). Please cite the original publication. Further, Fan et al., simplifies this formula for k=1, yet claims they use this formula to find an all-SNP F statistic (i.e., “where R2 is the proportion of variance in the exposure explained by the SNPs”). In the following sentence, they make an alternate claim (“we excluded any SNPs with an F-statistic below 10), suggesting they used the formula for one SNP at a time. Please be consistent in your explanation (i.e. from your comments it appears you calculated a SNP-specific F statistic, please update the sentence “where R2 is the proportion of variance in the exposure explained by the SNPs” to “where R2 is the proportion of variance in the exposure explained by the SNP”. Also, please explain how you calculated R2.

Response: Thank you for your valuable feedback. We have carefully revised our manuscript to address your concerns regarding the citation of the F statistic formula and the consistency in our explanation of calculating SNP-specific variance explained (R^2). We have now correctly cited the original publication by Burgess et al., 2011, for the F statistic formula and clarified our approach to calculating the SNP-specific F statistic, ensuring our methodology is accurately represented and consistent throughout the text. We believe these revisions address your comments and enhance the manuscript's clarity and accuracy.

4. You claim to have added “Output from UK Biobank - stratified by sex” – however I cannot find this in text.

Response: I appreciate your attention to detail and realize my response to your earlier query may have been overly simplistic. While I believed that referencing the specific IEU IDs in Table 1 would suffice to address your concerns, I understand now that this approach may not have clearly conveyed the information regarding the "Output from UK Biobank - stratified by sex". Acknowledging this oversight, I have now made further amendments to the manuscript to explicitly include this information within the text, ensuring clarity on the sex-specific nature of the testosterone, estradiol (E2), and sex hormone-binding globulin (SHBG) data sourced from the UK Biobank (UKB). Thank you for guiding me to make these necessary clarifications. See lines 194-197 for details

Revised: 

Replace " Additionally, total testosterone (TT), sex hormone-binding globulin (SHBG), and oestradiol (E2) were all obtained from the UK Biobank (UKB)". "

Revised to Additionally, total testosterone (TT), sex hormone-binding globulin (SHBG), and oestradiol (E2) datasets, stratified by sex to ensure specificity to female participants, were all obtained from the UK Biobank (UKB).

5. You have not ammended the manuscript text in response to my comment on the two types of horizontal pleiotropy. Additionally, the CAUSE paper indicates the two types of horizontal pleiotropy are (1) correlated pleiotropy and (2) uncorrelated pleiotropy, and that Egger regression and MR-PRESSO both address uncorrelated pleiotropy (Morrison et al., 2020). In line 247 you mention correlated and uncorrelated horizontal pleiotropy. Please explain what these are. Further, if the two types of horizontal pleiotropy are the correlated and uncorrelated horizontal pleiotropy, the sentences containing the phrases (line 246-247) “The CAUSE methodology effectively manages both correlated and uncorrelated horizontal pleiotropy” and (line 250-251) “CAUSE analysis addresses both types of horizontal pleiotropy,” are repetitive and thus could be combined into the one sentence.

Response: Thank you for your valuable feedback regarding the description of horizontal pleiotropy within our manuscript. Initially, I misunderstood your request as a simple inquiry for clarification, and therefore, did not amend the text to include a detailed explanation. I appreciate your guidance in highlighting this oversight. Following your advice, I have revised the manuscript to eliminate the repetitive statements and have now included a succinct yet comprehensive description of both correlated and uncorrelated horizontal pleiotropy, as distinguished in the CAUSE methodology per Morrison et al. (2020). This adjustment ensures a clearer understanding of the terms and reinforces the robustness of our analytical approach. See lines 253-263 for details

Revised: 

Replace " When unavoidable horizontal pleiotropy was detected by MR-PRESSO, the causal analysis using summary effect estimates (CAUSE) was applied[37]. The CAUSE methodology effectively manages both correlated and uncorrelated horizontal pleiotropy within genome-wide datasets. This approach utilizes genome-wide summary statistics to determine whether variant effects align with expected causal impacts. Unlike traditional MR methods, which rely on rigid assumptions, CAUSE analysis addresses both types of horizontal pleiotropy, thereby reducing the risk of false-positive results[37]. "

Revised to When unavoidable horizontal pleiotropy was detected by MR-PRESSO , the causal analysis using summary effect estimates (CAUSE) methodology was applied[37], effectively managing both correlated and uncorrelated horizontal pleiotropy within genome-wide datasets[37]. Correlated horizontal pleiotropy, wherein genetic variants influence multiple traits through shared heritable factors, presents a particular challenge by potentially mimicking causal relationships. CAUSE differentiates these effects from true causal relationships by utilizing genome-wide summary statistics, thus addressing both types of horizontal pleiotropy and significantly reducing the risk of false-positive results. This approach contrasts with Egger regression and MR-PRESSO, which primarily focus on uncorrelated horizontal pleiotropy by adjusting for pleiotropic effects that are independent of the genetic variant's association with the exposure. CAUSE thereby provides a comprehensive framework for analysis that surpasses traditional MR methods in flexibility and robustness.

6. Line 313 “Supplementary methods also yielded consistent findings.” – you have not addressed my question about what the supplementary methods are, nor amended the text based on this comment about what the supplementary methods are. Please explain what the particular methods are, and please include the reference to S3 table at the end of the sentence. For example: “Supplementary methods MR Egger, weighted median and CML also yielded consistent findings (S3 Table)”

Response: My sincerest apologies for the oversight in my previous responses regarding the supplementary methods used in our analysis. Initially, I did not fully grasp the specificity of your request. I have now corrected this misunderstanding and amended the manuscript accordingly to explicitly detail the supplementary methods employed—MR Egger, weighted median, and CML—and have included a reference to S3 Table at the end of the sentence to ensure clarity and comprehensiveness of the information provided. See lines 322-323 for details

Revised: 

Replace " Supplementary methods also yielded consistent findings "

Revised to Supplementary methods MR Egger, weighted median and CML also yielded consistent findings (S3 Table)

7. The updated title for table 2 detailed in the response to reviewers has not been implemented in text (line 347). Further, the MVMR results for all tested exposures should be available in the manuscript. Whilst I appreciate adding those results to the table will increase the length and thus the ease of reading, reporting all results is important for clarity and transparency. If you are concerned about Table 2 being too long, other solutions are possible, such as including the full set of results as a supplementary table instead of in text.

Response: Thank you for your continued guidance and constructive suggestions. Upon reflecting on your initial advice, I have now followed through by incorporating estimates for both exposures in the models, as you originally recommended. This adjustment has been made to ensure clarity and transparency in our findings, directly addressing your concerns. The revised Table 2, which now includes the estimates of both exposures within the models, has been updated accordingly in the manuscript. I appreciate your patience and understanding as we strive to improve the clarity and comprehensiveness of our work.

8. The response to the query about the CAUSE result (Query 10 within results) has not been implemented in text (line 334-335).

Response: I deeply apologize for the oversight in my previous submission, which led to the omission of the response to your query about the CAUSE result in the text you reviewed. The mistake occurred due to my error in not providing the correctly marked manuscript for your evaluation. I have now rectified this issue, and in the revised, marked manuscript, the response has been highlighted for clarity, specifically between lines 338-340. I regret any confusion this may have caused and appreciate your understanding and patience as we ensure all your concerns are addressed comprehensively.

9. Line 201 – the weighted median method is consistent when up to 50% of the information comes from invalid instrumental variables. Here you have indicated it is valid if >50 % of the information comes from invalid instrumental variables (through use of the term “a majority of the genetic variants”), which is not true. Please rephrase this sentence.

Response: Thank you for your critical observation regarding the description of the weighted median method's robustness in line 201 of our manuscript. I have now revised this sentence to accurately reflect that the weighted median method remains robust when up to 50% of the information comes from potentially invalid instrumental variables, providing a more precise description of its utility. I appreciate your guidance in ensuring the accuracy of our methodological descriptions. See lines 207-210 for details

Revised: 

Replace "The weighted median method becomes pivotal when a majority of the genetic variants are considered invalid"

Revised to The weighted median method remains robust when up to 50% of the information is derived from potentially invalid instrumental variables, allowing for reliable estimation

10. The manuscript would be further enhanced by explaining the difference between a mediator and a confounder.

Response: Thank you for your constructive feedback regarding the differentiation between mediators and confounders in our analysis. We have addressed this by clearly defining and distinguishing between these two critical concepts directly within the text. This clarification is now included following the initial discussion of our mediation and confounding factor findings, enhancing the manuscript's clarity and aiding in the accurate interpretation of our results. Specifically, we have elucidated the roles of diastolic blood pressure (DBP), systolic blood pressure (SBP), and triglycerides (TG) in the context of our study's findings, providing a clearer understanding of their respective roles as confounders and mediators in the relationship between serum lipids and endometriosis. We believe this addition satisfactorily addresses your concern and enhances the manuscript's contribution to the field. See lines 351-356 for details

---

## [Decision Letter · Decision Letter 2]

22 Mar 2024

Genetic association of serum lipids and lipid-modifying targets with endometriosis: Trans-ethnic Mendelian-randomization and Mediation Analysis

PONE-D-23-40604R2

Dear Dr. Fan,

We’re pleased to inform you that your manuscript has been judged scientifically suitable for publication and will be formally accepted for publication once it meets all outstanding technical requirements.

Kind regards,

Brett Mckinnon

Academic Editor

PLOS ONE

Additional Editor Comments (optional):

The authors have now satisfactorily addressed all of the reviewers comments. Congratulations on the interesting study.

Reviewers' comments:

Reviewer's Responses to Questions

**Comments to the Author**

1. If the authors have adequately addressed your comments raised in a previous round of review and you feel that this manuscript is now acceptable for publication, you may indicate that here to bypass the “Comments to the Author” section, enter your conflict of interest statement in the “Confidential to Editor” section, and submit your "Accept" recommendation.

Reviewer #1: All comments have been addressed

2. Is the manuscript technically sound, and do the data support the conclusions?

Reviewer #1: Yes

3. Has the statistical analysis been performed appropriately and rigorously? 

Reviewer #1: Yes

4. Have the authors made all data underlying the findings in their manuscript fully available?

Reviewer #1: Yes

5. Is the manuscript presented in an intelligible fashion and written in standard English?

Reviewer #1: Yes

6. Review Comments to the Author

Reviewer #1: All concerns have been appropriately addressed by the authors of this manuscript. It is ready for publication.

7. PLOS authors have the option to publish the peer review history of their article (what does this mean?). If published, this will include your full peer review and any attached files.

Reviewer #1: No

---

## [Editor Report · Acceptance letter]

22 May 2024

PONE-D-23-40604R2 

PLOS ONE

Dear Dr. Fan, 

I'm pleased to inform you that your manuscript has been deemed suitable for publication in PLOS ONE. Congratulations! Your manuscript is now being handed over to our production team.

Kind regards, 

on behalf of

Dr. Brett Mckinnon 

Academic Editor

PLOS ONE